# Development of Fractalkine-Targeted Nanofibers that Localize to Sites of Arterial Injury

**DOI:** 10.3390/nano10030420

**Published:** 2020-02-28

**Authors:** Hussein A. Kassam, David C. Gillis, Brooke R. Dandurand, Mark R. Karver, Nick D. Tsihlis, Samuel I. Stupp, Melina R. Kibbe

**Affiliations:** 1Department of Surgery, Center for Nanotechnology in Drug Delivery, University of North Carolina, Chapel Hill, NC 27599, USA; hussein_kassam@med.unc.edu (H.A.K.); david_gillis@med.unc.edu (D.C.G.); brooke_dandurand@med.unc.edu (B.R.D.); nick_tsihlis@med.unc.edu (N.D.T.); 2Simpson Querrey Institute, Northwestern University, Chicago, IL 60611, USA; mark.karver@northwestern.edu (M.R.K.); s-stupp@northwestern.edu (S.I.S.); 3Department of Chemistry, Northwestern University, Evanston, IL 60208, USA; 4Department of Materials Science and Engineering, Northwestern University, Evanston, IL 60208, USA; 5Department of Biomedical Engineering, Northwestern University, Evanston, IL 60208, USA; 6Department of Medicine, Northwestern University, Chicago, IL 60611, USA; 7Department of Biomedical Engineering, University of North Carolina, Chapel Hill, NC 27599, USA

**Keywords:** neointimal hyperplasia, arterial injury, cardiovascular disease prevention, fractalkine, nanofibers, targeted therapeutic, targeted delivery vehicle

## Abstract

Atherosclerosis is the leading cause of death and disability around the world, with current treatments limited by neointimal hyperplasia. Our goal was to synthesize, characterize, and evaluate an injectable, targeted nanomaterial that will specifically bind to the site of arterial injury. Our target protein is fractalkine, a chemokine involved in both neointimal hyperplasia and atherosclerosis. We showed increased fractalkine staining in rat carotid arteries 24 h following arterial injury and in the aorta of low-density lipoprotein receptor knockout (LDLR-/-) mice fed a high-fat diet for 16 weeks. Three peptide amphiphiles (PAs) were synthesized: fractalkine-targeted, scrambled, and a backbone PA. PAs were ≥90% pure on liquid chromatography/mass spectrometry (LCMS) and showed nanofiber formation on transmission electron microscopy (TEM). Rats systemically injected with fractalkine-targeted nanofibers 24 h after carotid artery balloon injury exhibited a 4.2-fold increase in fluorescence in the injured artery compared to the scrambled nanofiber (*p* < 0.001). No localization was observed in the non-injured artery or with the backbone nanofiber. Fluorescence of the fractalkine-targeted nanofiber increased in a dose dependent manner and was observed for up to 48 h. These data demonstrate the presence of fractalkine after arterial injury and the localization of our fractalkine-targeted nanofiber to the site of injury and serve as the foundation to develop this technology further.

## 1. Introduction

Cardiovascular disease is the leading cause of morbidity and mortality in industrialized nations and the largest cause of disability-adjusted life years globally [1]. Atherosclerosis is an inflammatory disease with the mainstay of treatment focusing on decreasing lipid burden and, for advanced disease, focusing on interventions such as angioplasty, stenting, or bypass grafting. Recently, the focus of lowering atherosclerotic disease burden has turned to methods of decreasing low-density lipoprotein-induced immune activation, albeit through in vitro studies and animal models. The notion of atherosclerosis as an immune-mediated disorder is based on evidence of immune activation and cytokine signaling within atherosclerotic lesions, and interference with immune signaling and inflammatory mediators inhibiting the pathogenesis of atheroma development in mouse models of atherosclerosis [2].

Cardiovascular disease interventions for the treatment of advanced atherosclerosis are limited by the development of neointimal hyperplasia, leading to a re-occlusive process that is multifactorial and involves inflammation, cellular proliferation and migration, and other processes [3,4,5]. Balloon dilation of the arterial wall provokes denudation of the endothelium, leading to exposure of vascular smooth muscle cells (VSMC) to serum containing pro-inflammatory cytokines, of which elevated levels are found in post-intervention patients [6]. Neointimal hyperplasia is mediated by a series of complex interactions of sustained autocrine and paracrine growth factor and cytokine expression on cells in the vascular wall, including VSMC and adventitial fibroblasts [7,8,9,10,11]. This cascade of events after arterial injury ultimately leads to narrowing of the arterial lumen and introduces further difficulty in re-intervention. Development of a therapy that effectively prevents atherosclerosis and inhibits the development of neointimal hyperplasia if vascular intervention is required is a significant unmet clinical need in cardiovascular medicine. Fractalkine is one such target involved in both disease processes. 

Fractalkine (CX3CL1) is a unique chemokine that exists as both a membrane-bound and soluble chemokine and is present on activated endothelium, dendritic cells, and VSMC [10]. It is anchored to vascular wall cells by an extended mucin stalk linked to a transmembrane domain and acts as an adhesion molecule for leukocytes (Appendix A) [11]. Fractalkine and its receptor (CX3CR1) regulate leukocyte adhesion and extravasation at the leukocyte–endothelial cell interface. More importantly, accumulating evidence suggests that fractalkine is involved in the pathogenesis of atherosclerosis, is highly expressed in early atherosclerotic lesions, and is expressed after arterial injury [12,13,14,15]. Mice with the ApoE and CX3CR1 genes deleted were shown to have less atheroma formation, and polymorphisms in these genes are associated with a significantly reduced risk of coronary artery disease [14,16]. Furthermore, inhibition of fractalkine by intraperitoneal injection of alpha-lipoic acid in rats after carotid artery balloon angioplasty has been shown to decrease neointimal hyperplasia development, mediated by the inhibition of the NF-kappaB pathway [15]. 

Current therapies approved by the United States Food and Drug Administration (FDA) to prevent neointimal hyperplasia are limited to drug-eluting stents and balloons, which deliver drugs locally to the site of injury [16,17]. A systemically delivered therapy that is targeted specifically to the site of interest in the vasculature would offer several advantages: (1) the opportunity for drug delivery without the long-term presence of a foreign body material; (2) the ability to deliver multiple doses of a drug and delivery of high local drug concentrations not possible with stents, which are limited by their drug carrying surface area; and (3) the avoidance of systemic side effects due to the targeted nature of the therapeutic. In this context, fractalkine is an ideal therapeutic target, as it would have targeting capability at the site of both the atheroma and neointimal hyperplasia development. The goal of the study was to develop a systemically delivered supramolecular nanostructure that targets the site of arterial injury. We hypothesized that a fractalkine-targeted nanofiber will localize to the site of arterial injury in a dose dependent fashion. 

## 2. Materials and Methods 

### 2.1. PA Synthesis and Labeling

Peptide amphiphile (PA) molecules were synthesized via standard 9-fluorenyl methoxycarbonyl (Fmoc) solid-phase peptide chemistry on Rink amide 4-methylbenzhydrylamine resin using a CEM Liberty Blue automated microwave peptide synthesizer (CEM Corporation; Matthews, NC, USA). Automated coupling reactions were performed using 4 equivalents of Fmoc-protected amino acid, 4 equivalents of *N*,*N’*-diisopropylcarbodiimide (DIC), and 8 equivalents of ethyl(hydroxyimino)cyanoacetate (Oxyma pure). Removal of the Fmoc groups was achieved with 20% 4-methylpiperidine in dimethylformamide (DMF). Peptides were cleaved from the resin using standard solutions of 95% trifluoroacetic acid (TFA), 2.5% water, and 2.5% triisopropylsilane (TIPS). Cysteine containing peptides included 3% 2,2′-(Ethylenedioxy)diethanethiol (DODT) and 92% TFA with the same TIPS/H_2_O mixture. Peptides were precipitated with cold ether and then purified by reverse-phase high-performance liquid chromatography on a Waters Prep150 (Milford, MA, USA) or Shimadzu Prominence high-performance liquid chromatograph (HPLC, West Chicago, IL, USA) using a water/acetonitrile (each containing 0.1% *v/v* TFA or 0.1% NH_4_OH) gradient. Eluting fractions containing the desired peptide were confirmed by mass spectrometry using an Agilent 6520 QTOF liquid chromatograph/mass spectrometer (LCMS; Santa Clara, CA, USA). Confirmed fractions were pooled and the acetonitrile was removed by rotary evaporation before freezing and lyophilization. Purity of lyophilized products was tested by LCMS. For PAs labeled with 5-carboxytetramethyrhodamine (TAMRA), the methtrityl (Mtt) protecting group was first removed from the lysine on-resin using 1% TFA in dichloromethane (DCM) with 5% TIS. After washing with DCM and DMF, TAMRA was then coupled to the now free epsilon amine of lysine using 1.2 equivalents of TAMRA, 1.2 equivalents of PyBOP (benzotriazol-1-yl-oxytripyrrolidinophosphonium hexafluorophosphate), and 8 equivalents of *N*,*N*-diisopropylethylamine for approximately 18 h. 

### 2.2. Conventional Transmission Electron Microscopy (TEM)

Conventional TEM images were taken on a FEI Tecnai T-12 TEM (ThermoFisher Scientific; Hillsboro, OR, USA) at 80 kV with an Orius^®^ 2k × 2k CCD camera (Gatan, Inc.; Pleasanton, CA, USA). PAs at 0.5 mg/mL in Hanks Balanced Salt Solution (HBSS) were prepared for TEM by negative staining. Briefly, 8 µL samples were incubated onto 400-mesh copper grids covered with a thin carbon film and previously treated with glow discharge. After 3 min, samples were stained with 2% uranyl acetate for 2–3 min and air-dried before imaging.

### 2.3. Circular Dichroism (CD) Spectroscopy

CD measurements were performed at a concentration of 0.125 mg/mL in HBSS using a Jasco J-815 CD spectrophotometer (Jasco Analytic Instruments; Easton, MD, USA) at 25 °C using a 1-mm path length demountable quartz cuvette. The far-ultraviolet (UV) spectral region (190–250 nm) was observed to determine if the assemblies contained secondary structure. Background subtraction of the HBSS buffer was performed. The data represent an average of two scans. Data points were taken every 0.2 nm with an analysis time per data point of 120 ms.

### 2.4. Study Approval

All animal procedures were performed in accordance with the Guide for the Care and Use of Laboratory Animals (National Institutes of Health Publication 85-23, 1996) and approved by the Animal Care and Use Committee at University of North Carolina Chapel Hill. The number of rats randomized to each treatment group (*n* = 3) was calculated using a power of 0.8, difference in means of 0.2, standard deviation of 0.1, and alpha of 0.05 using Stata/SE 15.1 (StataCorp LLC; College Station, TX, USA).

### 2.5. Animal Experiments

Adult male Sprague-Dawley rats weighing 300–350 g underwent carotid artery balloon injury as previously described [18]. After balloon injury, the arteriotomy was ligated and the animal was allowed to recover. Twenty-four hours after balloon injury, the different nanostructures were dissolved in 1 mL of 1× phosphate buffered saline (PBS) and administered via tail vein injection. Rats were euthanized at 5 h and 1, 2, 3, and 14 days post-injection, based on the experiment being conducted. The number of animals per treatment group for the different experiments were as follows: for localization of the nanofibers 5 h after injection, *n* = 4/treatment group; for localization duration study, *n* = 3/group; for concentration study, *n* = 3/treatment group.

Low-density lipoprotein receptor knockout (LDLR-/-) mice at 4 weeks of age were fed high fat diet consisting of 20% fat, 0.2% cholesterol, and 34% sucrose for 18 weeks. LDLR-/- mice develop severe hypercholesterolemia (>800 mg/dL) and hypertriglyceridemia (>300 mg/dL) as early as 2 weeks and atherosclerotic lesions after 12 weeks on high fat diet. At 16 weeks of age, aortic roots were harvested as previously described [19]. Tissue was harvested at 16 weeks of age to identify maximal atherosclerotic burden and later stages of atherosclerosis. 

### 2.6. Tissue Processing

#### 2.6.1. Tissue Harvest

After euthanasia via isoflurane overdose, bilateral thoracotomies were performed, followed by in situ perfusion with 1× PBS (250 mL) or until the liver appeared clear. Carotid arteries and viscera were harvested and then placed in 2% paraformaldehyde for 2 h at 4 °C, followed by 30% sucrose overnight at 4 °C. Tissue was processed as previously described [18]. Arterial tissue for light sheet microscopy was harvested after in situ perfusion with 1× PBS (250 mL) or until liver appeared clear and then placed in 4% paraformaldehyde at 4 °C for 2 h. Arterial tissue was then warmed in a 37 °C incubator for 30 min, removed from paraformaldehyde, and embedded in 1% agarose prepared in tri-acetate ethylenediaminetetraacetic acid EDTA buffer. Once agarose solidified, arteries were dehydrated by placing them in progressive methanol: DI H_2_O solutions, from 20% to 100%, at 20% intervals and 1 h incubation time per concentration. Tissue blocks were then removed, incubated in 66:33% (*v*/*v*) dichloromethane:methanol solution for 3 h at room temperature, and then transferred to 100% dichloromethane for 15 min. Lastly, arterial blocks were placed in 100% dibenzylether (DBE) until ready for imaging.

#### 2.6.2. Immunofluorescent Staining

Rat carotid arteries were harvested 24 h after injury and stained for CX3CL1 to determine presence of the protein. Primary antibody (rabbit polyclonal to CX3CL1, Abcam: ab25088) and secondary antibody (goat anti-rabbit IgG highly cross-adsorbed Alexa 647) dilutions were 1:200 and 1:3000, respectively. Aortic roots of mice fed a high-fat diet for 16 weeks were stained for CX3CR1 to determine presence of the protein in the atherosclerotic plaque. Primary antibody (rabbit polyclonal to CX3CR1, Abcam: ab8021) and secondary antibody (goat anti-rabbit IgG Alexa 555, Invitrogen or Alexa 647 highly cross-adsorbed, Invitrogen A32733) dilutions were 1:200 and 1:3000, respectively. Quantification of CX3CL1 staining data are presented as number of fluorescent pixels, which are the average of 5 images per animal, *n* = 3 animals per treatment group. Fluorescent imaging of CX3CR1 in arteries from atherosclerotic mice was obtained but not quantified due to low sample numbers. Results are expressed as mean number of fluorescent pixels ± the standard error of the mean (SEM).

Tissue sections from LDL KO mice were stained for CX3CR1 (Abcam ab8021) using primary dilution of 1:200 and secondary goat anti-rabbit Alexa 555 dilution of 1:3000 (Life Technologies A21425). Nuclear staining was carried out using DAPI at 1:500 dilution. Digital images were acquired as mentioned above.

### 2.7. Fluorescent Imaging

Carotid arteries harvested at respective time points underwent fluorescent imaging. Digital images were acquired using a Zeiss Axio Imager.A2 microscope (Hallbergmoos, Germany) with a 20× objective, HE Cy3 filter (Zeiss filter #43), using excitation and emission wavelengths of 550–575 nm and 605–670 nm, respectively, to assess presence of TAMRA-labeled PA fluorescence and immunohistofluorescence of stained arteries. The green fluorescent protein filter (Zeiss filter #38) using excitation and emission wavelengths of 470–495 and 525–550 nm, respectively, was used to assess tissue autofluorescence. To assess localization of the nanostructures, the fluorescent pixels of the arterial cross-sections were quantified in high power field (10× objective) images using ImageJ software (v.1.51, National Institutes of Health; Bethesda, MD, USA). Quantification data are presented as number of fluorescent pixels, which are the average of 5 images per animal, *n* ≥ 3 animals per treatment group. Results are expressed as mean ± SEM. 

### 2.8. Light Sheet Fluorescent Microscopy Imaging

Imaging was performed using a LaVision BioTec Ultramicroscope II (LaVision BioTec GmbH; Bielefeld, Germany) as previously described [20], with the following modifications. Cleared agarose-artery blocks mounted in a custom sample holder were submerged in 100% DBE. Artery images were acquired at 1.26× mag (0.63× zoom), using the three-light-sheet configuration with both left and right light sheets, with the horizontal focus centered in the middle of the field of view, a numerical aperture of 0.026 (sheet thickness at horizontal focus = 28 µm), and a light sheet width of 100%. Pixel size was 4.96 µm and Z-slice spacing was 14 µm. Two channels were imaged: arterial autofluorescence with 488 nm laser excitation and a Chroma ET525/50m emission filter and TAMRA channel with 561 nm laser excitation and a Chroma ET600/50m emission filter. Focus in both channels was ensured by use of the chromatic correction module on the instrument. Sample bleaching during initial imaging setup was minimized by use of low laser power and long exposure times. For image acquisition, higher laser power and shorter exposure times were used.

### 2.9. Statistical Analysis

JMP (SAS Institute, Inc.; Cary, NC, USA) was used to determine differences between groups depending on the study being conducted using an analysis of variance (ANOVA) followed by a Student’s t-test. Results are expressed as number of fluorescent pixels ± SEM. 

## 3. Results

### 3.1. Fractalkine Is Present in Injured Arteries and Atherosclerotic Plaques

To determine whether fractalkine was present in areas of arterial injury, rat carotid arteries harvested 24 h after balloon injury were stained for the fractalkine ligand CX3CL1 (Figure 1A). Positive staining was observed at the injured site compared to control non-injured artery, which was statistically significant (*p* = 0.028, Figure 1B). Staining was present in the media, where fractalkine has been shown to be upregulated in activated VSMC [15]. Arterial cross-sections of LDLR-/- mice fed a high-fat diet for 16 weeks exhibited positive staining for CX3CL1 in the atherosclerotic plaque and media (Appendix A).

### 3.2. Targeted and Non-Targeted PAs Form Nanofibers

To identify a unique targeting epitope that is present in both the atherosclerotic niche and neointimal hyperplasia, we reviewed the literature and found a peptide sequence that mimics a fragment of CX3CR1, the fractalkine receptor [21]. We chose one fractalkine-targeted sequence (SFPELDLENFEYDDSAEA) to synthesize into a peptide amphiphile (PA) C_16_-VVAAK[TAMRA]SFPELDLENFEYDDSAEA, where C_16_ a fatty acid palmitoyl group attached to the N-terminus of the peptide sequence. As controls, we synthesized the following PAs: a scrambled sequence with the same overall charge (C_16_-VVAAK[TAMRA]EYEDDFASNEFELPSDLA) and a backbone PA (C_16_-VVAAEEK[TAMRA]). Control PAs were synthesized to test the efficacy of the fractalkine-targeted sequence and the relationship of charge to targeting. For in vivo experiments, each PA was used individually at 100% for the nanofibers in comparison to previous studies in which targeted PAs were co-assembled with backbone filler PAs to produce different ratios of the targeting epitope [22,23,24]. TEM confirmed nanofiber formation for these different PAs (Figure 2 and Appendix A). 

### 3.3. Targeted and Non-Targeted Nanofibers Display Different Structures

The secondary structure of the fractalkine-targeted PA nanofibers was assessed by circular dichroism (CD) spectroscopy. The CD spectrum of the backbone PA nanofiber exhibits beta-sheet characteristics with a maxima around 200 nm and minima at 220 nm, the scrambled sequence PA nanofiber spectra suggests a random coil conformation with a strong minima at 198 nm, and the fractalkine-targeted PA nanofiber shows alpha-helical characteristics with minima at 202 and 216 nm (Appendix A). The LCMS analysis of the fractalkine-targeted, scrambled, and backbone nanofibers showed ≥90% purity and the expected masses at 3209, 3209, and 1393 *m*/*z*, respectively (Appendix A). 

### 3.4. Fractalkine-Targeted PA Nanofibers Localize to Injured Vasculature

The rat carotid artery balloon injury model was used to investigate the localization of the fractalkine-targeted, scrambled, and backbone nanofibers. PA molecules were labeled with the fluorescent dye TAMRA for visualization in vivo. After injection of the fluorescently labeled nanofibers, images of arterial cross sections showed that localization of the nanofiber occurred throughout the media at the injured site (Figure 3A). In all experiments, no signal was observed in the uninjured contralateral artery (Figure 3B). Fractalkine-targeted nanofiber exhibited a 4.2-fold increase in fluorescent signal compared to the scrambled nanofiber and 1400-fold increase compared to the backbone nanofiber (*p* < 0.001, Figure 3C).

### 3.5. Fractalkine-Targeted PA Nanofibers Exhibit a Dose-Dependent Relationship

The optimal dose of targeted nanofibers for subsequent in vivo studies was determined by injecting animals with a range of doses (0.5–5 mg). While the lowest dose appeared to show signal on arterial cross-sections, no quantifiable amount was observed (Figure 4). The lowest amount of targeted nanofiber injected that allowed for quantification was observed at 1.0 mg (Figure 4A). Quantification of the fluorescent signal on the arterial cross-sections revealed a statistically significant increase in fluorescent signal with injection of 5.0 mg versus 0.5, 1.0, and 2.5 mg (Figure 4B, *p* < 0.05). No statistically significant difference was observed between 1.0 and 0.5 mg (*p* = 0.053), 2.5 and 1.0 mg (*p* = 0.78), and 2.5 and 0.5 mg (*p* = 0.37), but there was a trend toward an increase in fluorescent pixels (Figure 4B).

### 3.6. Fractalkine-Targeted PA Nanofiber Is Present for 2 Days

Duration of localization for in vivo studies was assessed by injecting the optimal dose of 5.0 mg into animals and carrying out carotid artery harvest at 24, 48, and 72 h post-injection (*n* = 4/treatment group). Localization of the targeted nanofiber was observed at 24 and 48 h post-injection (Figure 5A). Quantification of the fluorescent signal on arterial cross-sections revealed a statistically significant increase at 24 and 48 h post-injection compared to control arteries (Figure 5B, *p* < 0.05). No fluorescent signal was detected at 72 h post-injection (Figure 5B).

## 4. Discussion

We describe here the presence of fractalkine in the arterial media after vascular injury and within the plaque in an atherosclerotic artery. Further, we describe the development and evaluation of a novel delivery vehicle designed to target fractalkine present in the artery after balloon injury. We show that our fractalkine-targeted PA nanofiber localizes to the area of arterial injury when administered by systemic intravenous injection. This targeted nanofiber can be administered over a range of dosages, with localization observed after injection of as little as 1.0 mg, and the targeted nanofiber remaining localized to the area of interest for up to two days. As such, these data represent the foundation for the development of a therapeutic that has potential to target both restenosis and atherosclerosis. 

We previously reported on a targeted therapy for the treatment of neointimal hyperplasia and atherosclerosis using a similar PA delivery vehicle, but with different targeting sequences [23,24]. In these studies, PAs with amino acid sequences that interact with type IV collagen or ApoA1 were used to target the area of neointimal hyperplasia and atherosclerotic plaque, respectively. The fractalkine-targeted PA is novel given that the protein is present in both these disease models and therefore one PA can be used to target the pathologic area in both diseases. It has previously been reported that fractalkine is present in the arterial media after arterial injury, albeit this was only accomplished through the evaluation of the effects of alpha-lipoic acid administration after balloon angioplasty [15]. Moreover, the effects of fractalkine in the development of atherosclerosis have been extensively studied. A polymorphism of CX3CR1, thereby resulting in the presence but lack of function of the fractalkine receptor, leads to the inability for leukocyte attachment and confers a reduced risk of acute coronary events [14,25]. Deficiency of the fractalkine receptor in ApoE-/- animal models provides significant protection from both macrophage recruitment to the vessel wall and subsequent development of atherosclerotic lesions [12]. While all these studies are promising, no literature exists on the development of a delivery vehicle targeting fractalkine in both a neointimal hyperplasia and atherosclerotic model. Furthermore, no literature exists on targeting the fractalkine receptor itself as a potential therapy in these two disease models. 

Our fractalkine-targeted PA has great potential for clinical use in the prevention of neointimal hyperplasia development after arterial injury. The ability to target the area of injury via intravenous injection, to be re-dosed if necessary, and the apparent safety profile of PAs makes this a great delivery vehicle for therapeutic agents [26]. In our previous study, a single 5.0 mg dose of S-nitrosylated collagen PA (collagen-SNO PA) was given immediately after injury and the therapeutic effects were seen up to seven months later. Animals exhibited a 51% decrease in intima/media ratio and a continued 45% decrease in percent occlusion of the injured artery [26]. We are also working to better quantify the dose–response observed with the fractalkine-targeted PA by continuing to expand the use of light-sheet fluorescence microscopy in our lab, which will allow for more accurate volume and area determinations. The ability of our fractalkine-targeted PA to remain localized at the area of injury for up to 48 h after systemic delivery further enhances its therapeutic potential. Previous studies have investigated different therapies to decrease expression of the fractalkine receptor but none have looked at the receptor itself as a therapeutic target [15]. Just as proprotein convertase subtilisin/kexin type 9 (PCSK9) inhibitors can target the LDL receptor, we envision a similar therapeutic agent targeting fractalkine, thereby preventing leukocyte recruitment and adhesion. Moreover, the involvement of fractalkine in both disease processes is due to the recruitment of inflammatory cells at the area of disease. By blocking the receptor and delivering a therapeutic agent with anti-inflammatory activity, we would anticipate a multiplicative therapeutic effect. Secondly, the specificity of our fractalkine-targeted PA would bypass any systemic side effects, contrary to those seen with statins and other medications, when used for primary and secondary prevention of atherosclerosis. Perhaps most importantly, the long-term therapeutic effects seen in our collagen-targeted SNO PA may translate to our fractalkine-targeted therapeutic PA. 

There are several limitations of our current study. First, our scrambled PA nanofiber exhibits a small degree of localization in the arterial media. This nanofiber possesses the same net charge (−7 at physiological pH) as the fractalkine-targeted nanofiber, and this charge may be the reason for the fluorescence that is observed in the injured artery. Preliminary studies conducted in the same animal model and injecting a nonsensical, negatively-charged (−7) nanofiber also exhibited a small amount of fluorescence (data not shown), indicating that both the charge and the fractalkine-targeted sequence are required for robust localization to the injured site. Second, while fractalkine is present in atherosclerotic arteries and is involved in the development of atherosclerosis, we have not tested our fractalkine-targeted PA in an atherosclerotic model, and we may not observe similar localization as in the arterial injury model. Third, while we have not determined the critical aggregation concentration for the fractalkine-targeted PA, we have previously shown that similar PA nanofibers can form in solutions containing serum [23]. 

Fourth, we did not determine the biodistribution of the fractalkine-targeted PA, although we can qualitatively say that the PA traffics to the typical organs of metabolism (liver and kidney). We have seen this previously with other PAs, and did not observe long-term adverse effects on a variety of biochemical markers (blood count, coagulation factors, liver enzymes, etc.) [26]. Biodistribution is the subject of ongoing research in the lab, and we have recently completed a large pharmacokinetic and biodistribution study of a targeted PA and observed a three-compartment model with rapid clearance from the intravascular space (manuscript in preparation). Of note, the fractalkine-targeted peptide sequence alone would likely be too small to target the luminal surface of the blood vessel, since the fatty acid tail of the amphiphile would not be present to encourage formation of long fibers (Figure 2B). The surface of the PA is also much larger, allowing for more epitope exposure, as well as margination of the long fibers to the low-flow regions near the vessel wall. Inclusion of the acyl tail allows for the ability to incorporate a therapeutic in the hydrophobic core of the nanofibers or covalently attach a therapeutic to a PA that could co-assemble with fractalkine-targeted PA. This would not be possible with the fractalkine-targeted peptide sequence alone.

Lastly, we have not tested the therapeutic effects of our fractalkine-targeted PA. Attaching a therapeutic agent may change the overall structure of our fractalkine-targeted PA that could lead to a lack of localization or, given the long targeting sequence, the inability to attach a therapeutic agent. In the latter case, we can attempt co-assembly of our fractalkine-targeted PA with a therapeutic PA attached to a backbone structure, thereby allowing for development of a therapeutic fractalkine-targeted nanofiber, as done in our previous work [21]. Optimizing the ratio of fractalkine-targeted to backbone PA in the supramolecular nanostructure that will still allow for targeting and drug delivery is ongoing in our lab. Future work will also include verifying the localization of the fractalkine-targeted PA in a CX3CR1-/- rat model, although this must be genetically engineered, as it does not currently exist. Finally, while direct head-to-head comparisons are difficult due to changes in backbone, epitope, charge, fluorophore, and time of PA administration, work to compare and contrast the properties of the various PAs our lab has used over the years will be the subject of future studies.

## 5. Conclusions

We showed that our fractalkine-targeted PA nanofiber binds to the site of arterial injury in a dose-dependent fashion and remains bound for two days, making this a promising drug delivery vehicle to prevent neointimal hyperplasia after arterial injury. Furthermore, the presence of fractalkine in atherosclerotic plaques and involvement in atherosclerosis makes our fractalkine-targeted PA nanofiber a unique avenue for investigation to prevent restenosis in the setting of atherosclerosis. 

## Figures and Tables

**Figure 1 nanomaterials-10-00420-f001:**
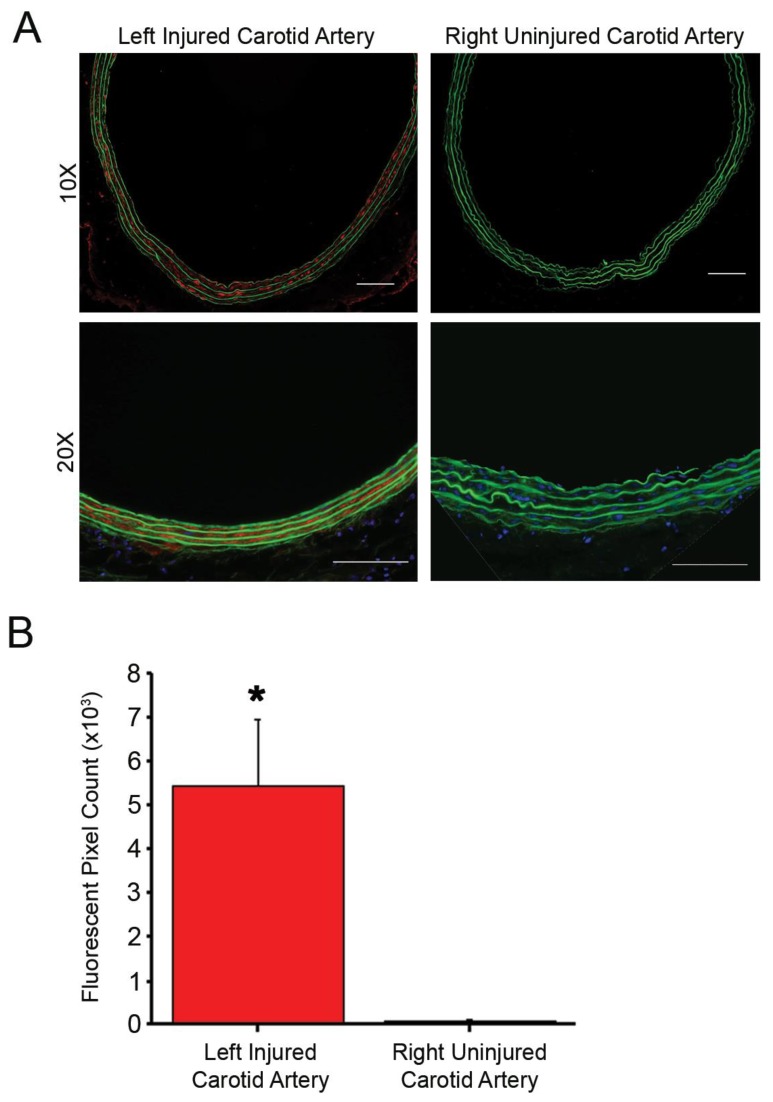
Immunofluorescent staining of fractalkine is present in left injured carotid artery with no fluorescent staining in the control uninjured right carotid artery. (**A**) Fluorescent microscopy of left injured and right uninjured carotid artery stained for CX3CL1. Green = autofluorescence of arterial lamina and red = CX3CL1 staining. Magnification, 20×; *n* = 3/treatment group; scale bar = 100 µm. (**B**) Quantification of fluorescence demonstrates the elevated expression of fractalkine in the injured artery compared to control uninjured artery, * *p* < 0.05.

**Figure 2 nanomaterials-10-00420-f002:**
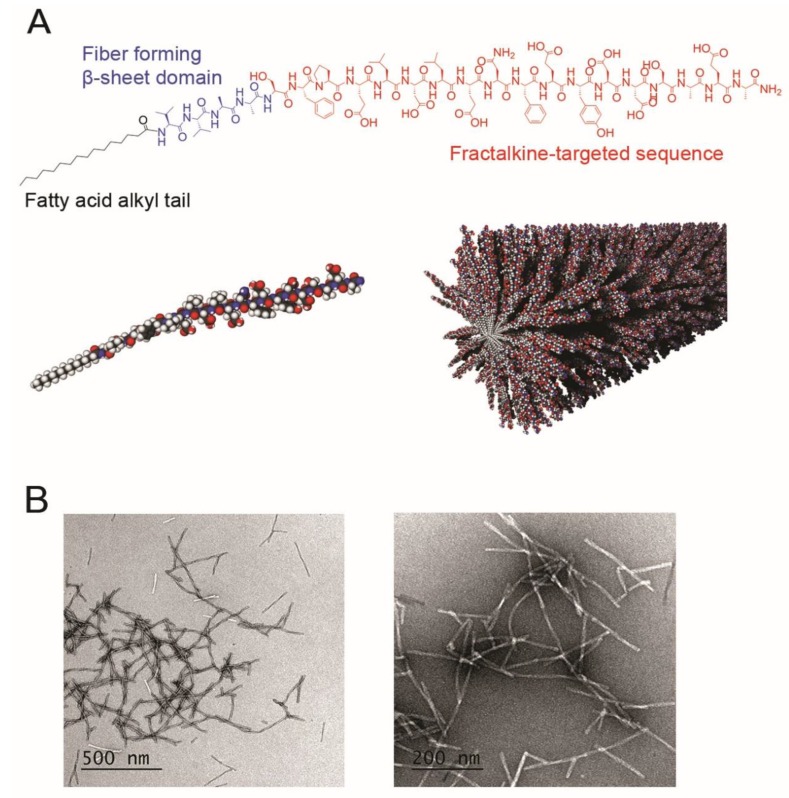
(**A**) Chemical structure of the fractalkine-targeted peptide amphiphile. Red = fractalkine-targeted amino acid sequence, blue = beta-sheet forming sequence, and black = fatty acid alkyl tail. Three-dimensional modeling of the fractalkine-targeted peptide amphiphile is shown below the chemical structure. (**B**) Conventional transmission electron microscopy (TEM) images of fractalkine-targeted nanofibers in positive (left) and negative (right) birefringent exhibiting nanofiber formation.

**Figure 3 nanomaterials-10-00420-f003:**
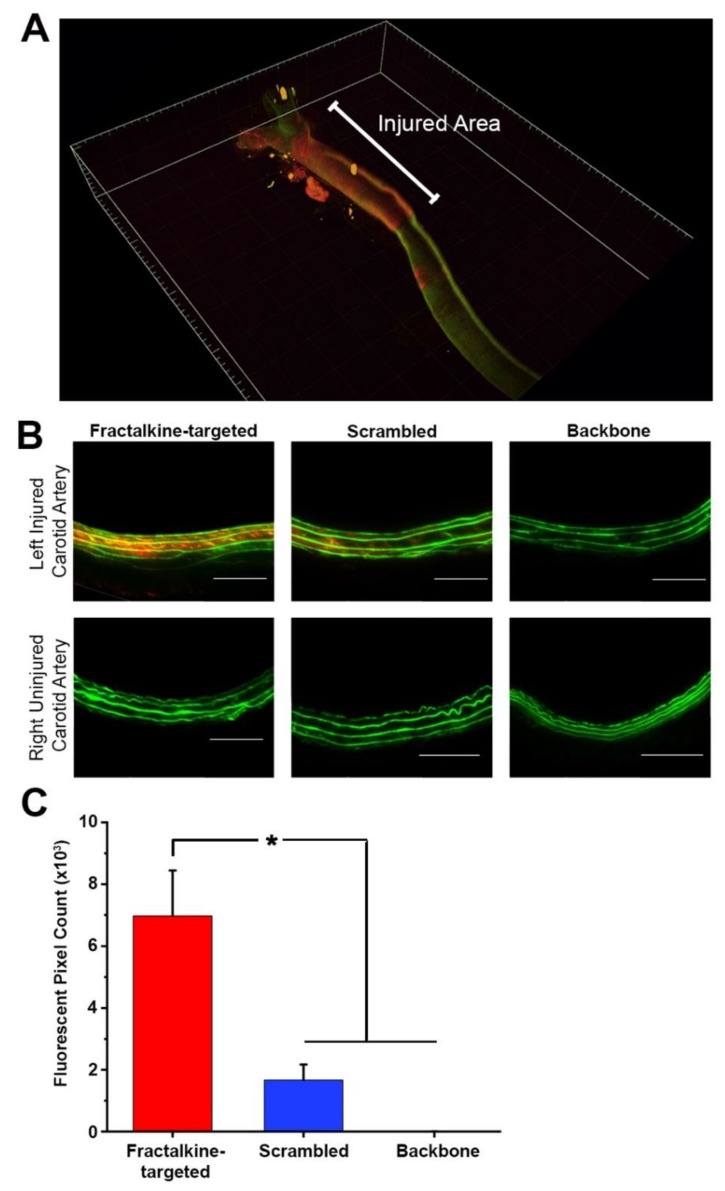
Fractalkine-targeted nanofiber localizes to the medial surface of left injured carotid artery and in greater amounts compared to control peptide amphiphiles, without localizing to the uninjured right carotid artery. (**A**) Three-dimensional light-sheet fluorescence microscopy exhibiting localization of fractalkine-targeted nanofiber to the injured area **(B**) Fluorescent microscopy of left injured and right uninjured carotid arteries show localization of fractalkine-targeted nanofiber to injured site (red), minimal localization of the scrambled nanofiber, and no localization of the backbone nanofibers to the injured site. Green = autofluorescence of arterial lamina and red = TAMRA-tagged nanofiber. Magnification, 20×; *n* = 4/treatment group; scale bar = 100 µm. (**C**) Quantification of fluorescence demonstrates localization of fractalkine-targeted PA to the injured artery compared to control nanofibers at the injured site, * *p* < 0.001.

**Figure 4 nanomaterials-10-00420-f004:**
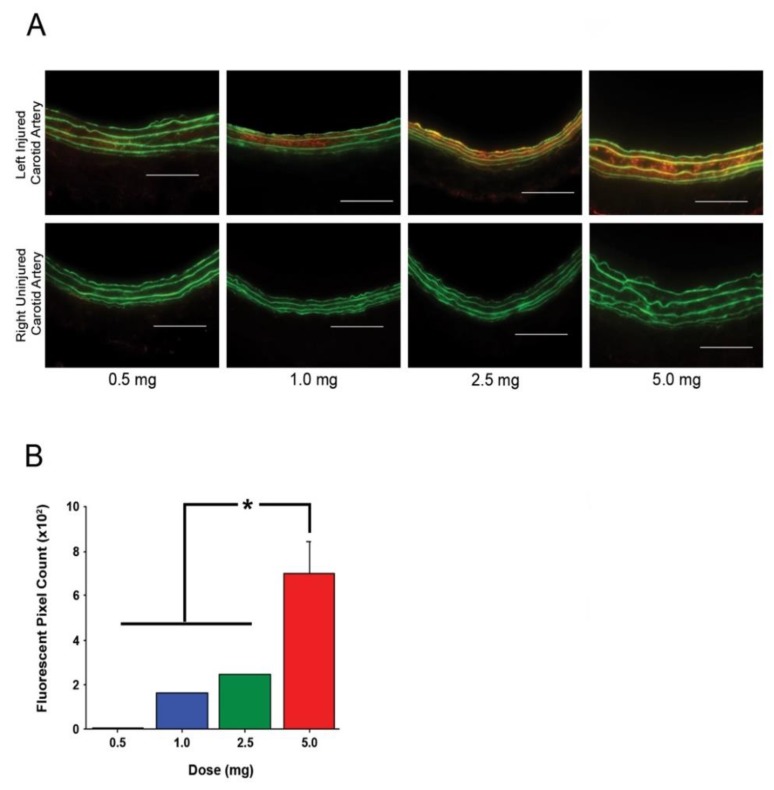
Fractalkine-targeted nanofiber exhibits dose-dependent localization at the left injured carotid artery without localizing to the right uninjured carotid artery. (**A**) Fluorescent signal of fractalkine-targeted nanofiber to injured carotid artery is increased with increasing doses of peptide amphiphile injection. Green = autofluorescence of arterial lamina and red = TAMRA-tagged nanofiber. Magnification, 20×; *n* = 3; scale bar = 100 µm. (**B**) Quantification of fluorescence demonstrates localization of fractalkine-targeted PA to the injured artery increases with increasing amounts of PA injection, * *p* < 0.05.

**Figure 5 nanomaterials-10-00420-f005:**
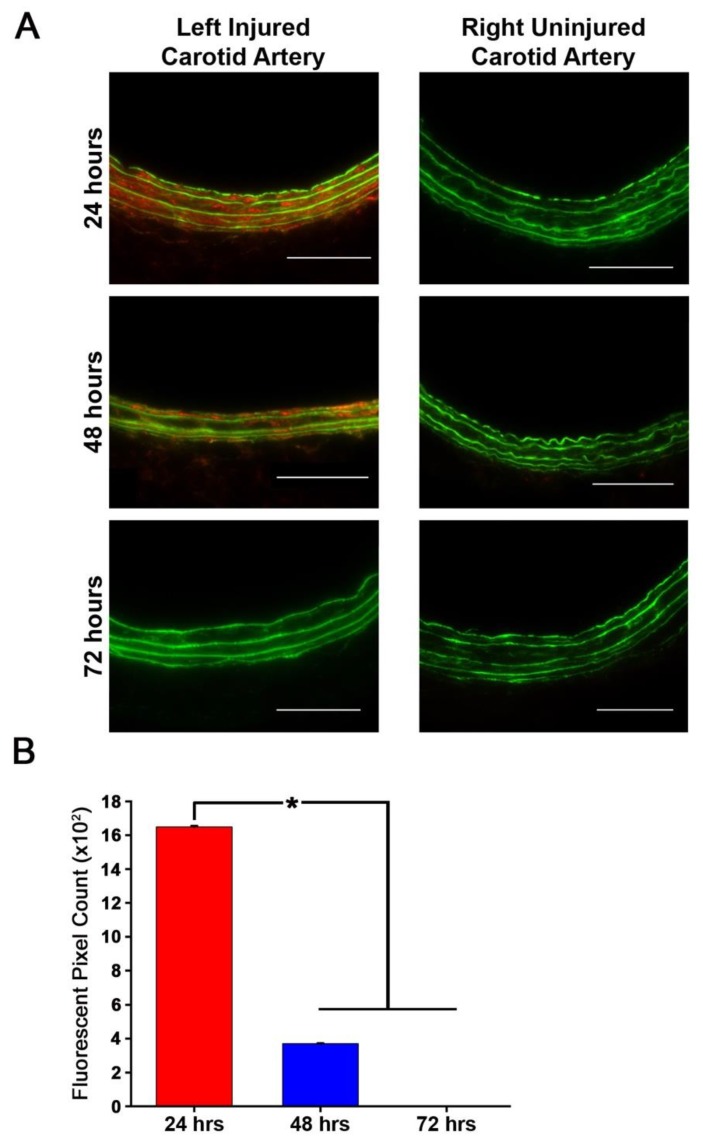
Fractalkine-targeted nanofiber remains localized to the left injured carotid artery for 24 and 48 h without localizing to the right uninjured carotid artery. (**A**) Fluorescent microscopy of fractalkine-targeted nanofiber in injured carotid arteries exhibits localization to the medial surface of the left injured carotid artery for up to 48 h with no remaining nanofibers at 72 h and no localization in the right uninjured carotid artery. Green = autofluorescence of arterial lamina and red = TAMRA-tagged nanofiber. Magnification, 20×; *n* = 3; scale bar = 100 µm. (**B**) Quantification of fluorescence demonstrates localization of fractalkine-targeted nanofiber to the injured artery at 24 and 48 h post injection and no remaining quantifiable fluorescence at 72 h, * *p* < 0.05.

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
