# Peer review of "Development of Fractalkine-Targeted Nanofibers that Localize to Sites of Arterial Injury"

_nanomaterials, 2020, doi:10.3390/nano10030420_

Round 1
Reviewer 1 Report
The manuscript is well written and scientifically sound. The subject is of the research is of general interest and could attract broad reader interest. The animal experiment has highlighted the importance of this research beyond academic research. There are a few questions that arise from the design of the research.
The first is the aggregation issue. The supramolecular assemblies presented in this manuscript tend to aggregate which would limit their bioavailability and could compromise the pharmacokinetics. A detailed evaluation (size exclusion chromatography- DLS etc.) of the aggregate formation is necessary for the molecular assemblies in solutions in the presence and absence of BSA in order to understand their behavior in physiological liquids. These experiments should be presented in order to complete the characterization of the synthesized nanoassemblies. What is the difference in the targeting efficacy if only a fractalkine-targeted peptide sequence would be used? The second question is for the animal model evaluation. The researchers have shown the nice targeting possibility of the described nanoassemblies. Nevertheless, the information on their systemic distribution (bioavailability) in the animal organs is missing. The accumulation of such aggregates in the liver and kidneys as well as in the capillary periphery is important to understand if these nanoassemblies have any perspective in medical application. The ethical approval of animal research should be attached to the supplemental data.
Author Response
Please see the attachment.
Response to Reviewers
Reviewer 1
The first is the aggregation issue. The supramolecular assemblies presented in this manuscript tend to aggregate which would limit their bioavailability and could compromise the pharmacokinetics. A detailed evaluation (size exclusion chromatography- DLS etc.) of the aggregate formation is necessary for the molecular assemblies in solutions in the presence and absence of BSA in order to understand their behavior in physiological liquids. These experiments should be presented in order to complete the characterization of the synthesized nanoassemblies.
We agree that pharmacokinetics can be compromised by the aggregation of supramolecular nanostructures, which is why our lab is thoroughly investigating the pharmacokinetic and pharmacodynamics properties of a similar peptide amphiphile (PA) that will address the reviewer’s concerns about bioavailability. However, the in vitro environment used for the TEM does not mimic the complex in vivo environment. While we have not determined the critical aggregation concentration of our fractalkine-targeted PA in the presence and absence of BSA, we have determined and published the critical aggregation concentration of a very similar PA and previously shown that PAs are able to form fibers in BSA-containing solutions. We have included this information as well as the references in the limitations section of the discussion (Lines 363-372). Our current results show targeting of the injured site in live animal models, which this takes into account the presence of physiological concentrations of serum proteins in addition to the complex interaction between other proteins, organ systems, and physiologic changes. We feel these in vivo data are more revealing about the ability of our therapy to target injured areas.
What is the difference in the targeting efficacy if only a fractalkine-targeted peptide sequence would be used?
The fractalkine-targeted peptide sequence alone would likely be too small to target the luminal surface of the blood vessel, since the fatty acid tail of the amphiphile would not be present to encourage formation of the long fibers seen in Figure 2B. The surface of the PA is also much larger, allowing for more epitope exposure, as well as margination of the long fibers to the low-flow regions near the vessel wall. By incorporating the acyl tail, this allows us the ability to incorporate a therapeutic in the hydrophobic core of the nanofibers, or covalently attach a therapeutic to a PA that we could co-assemble with fractalkine-targeted PA. We could not do that with the fractalkine-targeted peptide sequence alone, so that would not be an approach we could pursue, given our overall goal. To the reviewer’s point, we are currently exploring optimizing the ratio of fractalkine-targeted to backbone PA in the supramolecular nanostructure that will still allow for targeting and drug delivery. We have revised the discussion section to include this information (Lines 372-380, and lines 387-389).
The second question is for the animal model evaluation. The researchers have shown the nice targeting possibility of the described nanoassemblies. Nevertheless, the information on their systemic distribution (bioavailability) in the animal organs is missing. The accumulation of such aggregates in the liver and kidneys as well as in the capillary periphery is important to understand if these nanoassemblies have any perspective in medical application.
We agree with the reviewer that this is a very important area of investigation. As such, we have previously investigated and reported on the presence of targeted PA nanofibers in other organ systems and the physiologic effects, safety, and biocompatibility of the PA. Based on these data, presence of fluorescence in the liver and kidney would be expected, given these are likely the organs of metabolism and excretion of the breakdown products of the PA, respectively. Furthermore, blood work analysis after injection of targeted and backbone PA has revealed the safety and biocompatibility of the PA nanofiber. Lastly, in collaboration with a pharmacologist, we have recently completed a large pharmacokinetic and biodistribution study of a targeted PA and observed a 3-compartment model with rapid clearance from the intravascular space. These prior publications as well as the recent unpublished pharmacokinetic and biodistribution data have been included in the discussion section, along with the appropriate references (Lines 366-392).
The ethical approval of animal research should be attached to the supplemental data.
As noted in Section 2.4, the animal study was approved by the Animal Care and Use Committee at UNC. This information is included in the methods section of the manuscript (Lines 121-123).

Reviewer 2 Report
I found this research on the "Development of Fractalkine-targeted Nanofibers that Localize to Sites of Arterial Injury" appropriately designed and executed however I have the following questions and concerns regarding this work:
As authors also pointed out through the manuscript, several amphiphilic peptides have been developed by Stupp's group to target several diseases including arterial injury and neointimal hyperplasia. How efficient does this particular peptide function compared to previous ones? In fact, a comparison study (or a literature review on previous studies) can benefit this work. Circular dichroism spectroscopy has been a widely used technique in structural biology for determining the secondary structure of peptides. Authors have provided one supplementary figure (Supplemental 5 CD Fractalkine BB Scram data from 6-11-19.jpg) for presented work but not well discussed. Additional quantification evaluations on designed peptide (including half-life, integrity, etc) will greatly benefit the presented work. Image analysis on quantification of fluorescence signal for demonstration of localization of fractalkine-targeted PA to the injured site (figure 3, figure 4, and figure 5) can be furthered improved by normalizing the total signal per arterial surface area. This information can be captured by using a secondary stain in order to truly normalized the total quantified signal per surface area of the injured arterial wall (rather than magnification scale) in order to further quantify and assess the true difference among different groups (in particular for dosage study). Regarding in vivo study design, I recommend authors to include a study on CX3CR1 knock out as a negative control to further validate the efficiency of this peptide.
Reviewer 3 Report
Comments to the authors
Page 2 line 60 : references 12 and 13 are missing, they are not mentioned in the text.
Page 4 in the paragraph immunofluorescence staining : why are you using 2 different anti-CX3CL1 antibodies ?
Page 6 line 224 : put reference 22 in square brackets. In addition, references 20 and 21 are missing, they are not mentioned in the text, you stop at reference 19 on page 3 line 140.
Page 7 line 233 : put reference 23 in square brackets.
Page 8 paragraph 3.4 : Figures 3B and C are not quoted or commented on
In the figures it would be more readable with bars to have an estimate of the size of the arteries rather than putting X20.
Page 11 line 295 : add Figures 5A, B because on A, we can see as well that after 72 hours there is no more fluorescent signal
Discussion : all references cited must be put in square brackets (lines 316, 323, 326, 328, 335, 338, 341, 365)
In section 2.5 : you mention the use of knockout mice LDLR-/- but you don't show any results ???
In the Supplementary : please put legends under the different figures to clarify more.
Review legend for supplemental Figure 4A (page 13 line 388) corresponding to conventional electron microscopy.
Round 2
Reviewer 2 Report
I want to thank the authors to revise the manuscript according to the raised questions and commented on several suggestions and concerns regarding this work. After careful review of all feedbacks I recommend the revised manuscript to the editor for publication.
